# Predictive Analysis of Healthcare-Associated Blood Stream Infections in the Neonatal Intensive Care Unit Using Artificial Intelligence: A Single Center Study

**DOI:** 10.3390/ijerph19052498

**Published:** 2022-02-22

**Authors:** Emma Montella, Antonino Ferraro, Giancarlo Sperlì, Maria Triassi, Stefania Santini, Giovanni Improta

**Affiliations:** 1Department of Public Health, University of Naples “Federico”, 80125 Naples, Italy; emma.montella@unina.it (E.M.); maria.triassi@unina.it (M.T.); giovanni.improta@unina.it (G.I.); 2Department of Information Technology and Electrical Engineering, University of Naples “Federico”, Via Claudio 21, 80125 Naples, Italy; antonino.ferraro@unina.it (A.F.); stefania.santini@unina.it (S.S.); 3CINI-ITEM National Lab, Complesso Universitario di Monte S. Angelo Via Cinthia Edificio Centri Comuni, 80126 Naples, Italy; 4Interdepartmental Center for Research in Healthcare Management and Innovation in Healthcare (CIRMIS), University of Naples “Federico”, 80131 Naples, Italy

**Keywords:** statistical analysis, predictive analysis, healthcare-associated infection, healthcare

## Abstract

Background: Neonatal infections represent one of the six main types of healthcare-associated infections and have resulted in increasing mortality rates in recent years due to preterm births or problems arising from childbirth. Although advances in obstetrics and technologies have minimized the number of deaths related to birth, different challenges have emerged in identifying the main factors affecting mortality and morbidity. Dataset characterization: We investigated healthcare-associated infections in a cohort of 1203 patients at the level III Neonatal Intensive Care Unit (ICU) of the “Federico II” University Hospital in Naples from 2016 to 2020 (60 months). Methods: The present paper used statistical analyses and logistic regression to identify an association between healthcare-associated blood stream infection (HABSIs) and the available risk factors in neonates and prevent their spread. We designed a supervised approach to predict whether a patient suffered from HABSI using seven different artificial intelligence models. Results: We analyzed a cohort of 1203 patients and found that birthweight and central line catheterization days were the most important predictors of suffering from HABSI. Conclusions: Our statistical analyses showed that birthweight and central line catheterization days were significant predictors of suffering from HABSI. Patients suffering from HABSI had lower gestational age and birthweight, which led to longer hospitalization and umbilical and central line catheterization days than non-HABSI neonates. The predictive analysis achieved the highest Area Under Curve (AUC), accuracy and F1-macro score in the prediction of HABSIs using Logistic Regression (LR) and Multi-layer Perceptron (MLP) models, which better resolved the imbalanced dataset (65 infected and 1038 healthy).

## 1. Introduction

The European Centre for Disease Prevention and Control (ECDC) recognized neonatal infections as one of the six main types of healthcare-associated infections (HABSIs) in 2016 [1]. The first few months of neonatal life, especially the first month, are the main period of risk for child survival [2,3], whose mortality rate has increased approximately 50% due to preterm births or problems derived from childbirth. Although advances in obstetrics and technologies have minimized the number of birth-related problems, HABSIs pose different challenges in identifying the main factors affecting mortality and morbidity, which represent the main reason for mortality in children with a birth weight less than 1500 g in the Neonatal Intensive Care Unit (NICU) [4]. An increase in the incidence of HABSIs in NICUs was found in [5] due to the relationship between the quantity and invasiveness of surgical procedures, because these patients are immature from an immunological point of view [6,7,8].

The present paper performed a risk assessment using statistical analyses and logistic regression to reveal the main factors that affect neonates suffering from a healthcare-associated blood stream infection (HABSI) infection [9,10]. The definition of HABSI provided by the European Center Disease Control (ECDC) is the same as the CDC in Atlanta and it is based on clinical (e.g., fever) and/or microbiological (e.g., increased white blood cells and isolation of germs). The hospital was equipped with a manual that contained all of the prevention protocols for healthcare-related infections. Each protocol had a monitoring sheet for correct use, and incidence studies made it possible to measure the effectiveness of the correct application via the calculation of specific indicators. For example, the incidence rate of pneumonia was associated with assisted ventilation [11], and the incidence of urinary tract infections was associated with the use of a bladder catheter [12]. We further provided a predictive analysis of the different features in approximately 1203 neonates, 65 of who were diagnosed as suffering from an HABSI contracted as a healthcare-associated blood stream infection (HABSI) using seven different artificial intelligence models that are increasingly applied in the medical field [13,14,15]. Notably, the main novelties of our approach concerned the analysis of HABSIs from different points of view (predictive and statistical analyses) in a cohort of 1203 patients at the level III NICU of the “Federico II” University Hospital in Naples from 2016 to 2020 (60 months). Despite different efforts in the literature for predicting the risk of HABSIs and other adverse outcomes in the ICU, these studies primarily focused on the use of Support Vector Machine (SVM) [16,17] or divided patients into clusters [18] without considering more recent Artificial Intelligence approaches (e.g., XGB or Catboost) or the large set of features in our analyses. Notably, our approach achieved better classification performances in accuracy and Area Under Curve (AUC) [16,17].

This paper is organized as follows. The results are shown in Section 2, and discussed in Section 3. Some conclusions and future works are discussed in Section 5.

## 2. Materials and Methods

This section discusses the characterization of the study population, which was composed of 1203 infants who stayed longer than two days in the level III NICU of the “Federico II” University Hospital in Naples from 2016 to 2020. These patients were continuously monitored by healthcare staff in all hospital ICUs and suffered from healthcare associated blood stream infection (HABSI). In particular, HABSIs were observed in the next two days after the date of admission at the NICU on the basis of the Disease Control and Prevention directive for neonatal acute care settings [19]. The data were collected in a prospective manner by including all patients who were admitted to the NICU from a single data source, QuaniSDO (used for the computerization of hospital discharge forms). All hospitalized patients have a risk record from the moment of admission to discharge, in which the epidemiologically important data for the study of hospital infections were collected. We did not consider patients with vertical or other infections due to childbirth in our analysis. In particular, our study did not consider childbirth-related or vertically transmitted infections because women who give birth are subjected to vaginal and rectal swabs for the research of bacteria related to infection, and the search for viruses is performed using blood.

Viral infections may be vertically transmitted from mother to child at different times, from “in utero” transmission, which occurs during the gestational age, perinatal transmission, which occurs during delivery, and postnatal transmission, which is generally a consequence of breastfeeding. Mother-to-child transmission, which may occur after primary, recurrent or chronic maternal infection, is potentially harmful to the fetus or the newborn because it may result in miscarriage, fetal death, congenital anomalies, intrauterine growth restriction or severe neonatal disease. Some risk factors may affect the rate of mother-to-child transmission, such as the presence of other viral infections, maternal viral load, type of infection (primary versus recurrent), obstetrical procedures (prolonged rupture of membranes and mode of delivery), socioeconomic conditions and breastfeeding. For some vertically transmitted viruses, interventions are available to prevent mother-to-child transmission, such as vaccines, passive immunization, and antiviral drugs. Perinatal and postnatal infections may be prevented with elective caesarean delivery and avoidance of breastfeeding [20]. Fetal infection also occurs from the aspiration of infected amniotic fluid. Some viruses are present in genital secretions or the blood (Herpes Virus Simplex (HSV), citomegalovirus (CMV), hepatitis B virus (HBV), Hepatitis C Virus (HCV), and Human Papillomavirus Infection (HPI)). The newborn may be colonized at birth during passage through the birth canal. The most commonly involved microorganisms are Gram + cocci (streptococci and staphylococci), Gram cocci (*Neisseria*), Gram enteric bacilli (*Escherichia coli*, *Proteus* sp., *Klebsiella* sp., *Pseudomonas* sp., *Salmonella* and *Shigella*), anaerobic bacteria (*Mushrooms*), chlamydia (*Protozoa*) (Trichomonas vaginalis and Toxoplasma gondii), and mycoplasmas (Viruses). However, the association with neonatal disease was significant only for Group A and Group B streptococci, Escherichia coli, Neisseria gonorrhoeae, CMV, HSV type II, Candida albicans, and Chlamydia trachomatis.

We consider different clinical information (e.g., birthweight (in grams), gestational age (in weeks), sex, length of total hospital stay and invasive device exposure (days of umbilical and central line catheterization) for each patient. The patients studied were not stratified by disease severity, but the database used reported the score of American Society of Anesthesiologists (ASA). We had mortality data for both groups, but they were not included in the processed dataset. Notably, patients informed consent and local Ethical Committee authorization were not required, and all of the data came from HAI surveillance, which is regulated by the Regional Health Authority (https://www.aslsalerno.it/documents/20181/147671/PianoRegionaleICA.pdf/67a62ec9-ade9-4552-b389-5b8dabe72e6a, accessed on: 5 January 2022).

The mission and activities of A.O. U of University of Naples Federico II, which consists of a multiblock building complex, and an organizational model that provides Departments of Integrated Activity (DAI), ensures the integrated exercise of care, teaching and research and the unitary management of economic, human and instrumental resources. Twice a year (November and March), the Federico II University Hospital of Naples participates in the prevalence study of HAI organized by the ECDC, which allows the comparison of the data on the European territory and the evaluation of its trend over time to highlight any critical changes. In high-risk areas, such as NICUs, incidence studies are performed with staff who collect data in the risk records processed for each patient from admission to discharge. The data allow the preparation of reports on a quarterly basis subject to clinical audits with medical and nursing staff working in the affected areas. Training courses are organized at least three times per year for all health personnel, with compulsory attendance as part of working hours.

We performed our analysis using the chi-squared test, Fisher’s exact test and Kruskal-Wallis test, as appropriate, but the relationships between the dependent variable (HABSIs) and the different risk factors under study (e.g., sex, length of total hospital stay or gestational age) were investigated using logistic regression. Notably, we considered associations significant when the *p*-value was less than the threshold value 0.05. We further provided a predictive analysis by splitting the cohort of patients in 80% and 20% to train and test different artificial intelligence models (Support Vector Machine (SVC), CATBOOST, XGBoost (XGB), Ranger Forest Classifier (RFC), Multi-Layer Perceptron (MLP), Random Forest (RF) and Logistic Regression (LR)) in predicting whether a patient suffered from HABSI. Samples of two classes in the training set were balanced using the data augmentation method. To optimize the parameters of each model, we divided the training set (the 80% of patients) into two groups: the former set was used to train the chosen model using a specific set of parameters, which are described in Table 1; and the latter set validated the model’s performance.

Finally, the analysis was performed on Google Colab (https://colab.research.google.com/, accessed on: 5 January 2022), a Platform As A Service (PAAS) platform provided by Google with one single core hyperthreaded Xeon Processor @2.2 Ghz, 12 GB of RAM and a Tesla T4 GPU, using Python 3.6 with the scipy (https://scipy.org/, accessed on: 5 January 2022), statsmodels (https://www.statsmodels.org/, accessed on: 5 January 2022), scikit-learn (https://scikit-learn.org/, accessed on: 5 January 2022), catboost.ai (https://catboost.ai/, accessed on: 5 January 2022) and skranger (https://pypi.org/project/skranger/, accessed on: 5 January 2022) libraries.

## 3. Results

### 3.1. Statistical Analysis

This section discusses the statistical analyses used to investigate possible correlations between different risk factors (as independent variables) and the possible occurrence of HABSI disease (as dependent variable). Notably, the population under study included all of the neonates who stayed at least two days at the NICU of the Federico II from 2016 to 2020 (60 months) for a total of 1203 neonates, and 65 suffered from HABSI (Table 2). Neonates affected by HABSI had a shorter gestational age and lower birthweight than non-HABSI neonates. The total length of hospitalization and umbilical line and central line catheterization days were longer in HABSI patients.

Table 3 provides the multivariate analysis results, which confirmed that the significant predictors of suffering from HABSI were only birthweight and central line catheterization days.

### 3.2. Predictive Analysis

This section provides a predictive analysis on the basis of different risk factors that were first normalized and successively processed to predict whether a patient suffered from HABSI. We evaluated our results in terms of accuracy, AUC, and F1-score, whose formal definitions are provided in Equations (Equation 1)–(Equation 3) and Macro-F1 (Equation 4). In particular, the AUC was computed as the probability that a classifier would rank a randomly chosen positive example higher than a randomly chosen negative example).
(1)$$\mathrm{Accuracy}=\frac{\mathrm{Number}\;\mathrm{of}\;\mathrm{correct}\;\mathrm{predictions}}{\mathrm{Total}\;\mathrm{number}\;\mathrm{of}\;\mathrm{predictions}}$$
(2)$$\mathrm{AUC}=P(\mathrm{score}\left(x^{+}\right)>\mathrm{score}\left(x^{-}\right))$$
(3)$$F1\text{-}\mathrm{score}=2*\frac{\left(\mathrm{precision}*\mathrm{recall}\right)}{\left(\mathrm{precision}+\mathrm{recall}\right)}$$
(4)$$F1\text{-}\mathrm{score}=\frac{1}{N}\sum_{i=0}^{N}{F1\text{-}\mathrm{score}}_{i}\left(i\;\mathrm{class}\;\mathrm{label}\;\mathrm{and}\;N\;\mathrm{number}\;\mathrm{of}\;\mathrm{classes}\right).$$

Different Artificial Intelligence models (Support Vector Machine (SVC), CATBOOST, XGBoost (XGB), Ranger Forest Classifier (RFC), Multi-Layer Perceptron (MLP), Random Forest (RF) and Logistic Regression (LR)) were used to predict whether a patient suffered from HABSI. We analyzed a cohort of 1203 neonates who were divided into 80% and 20% for the training and test sets, respectively, using *train_test_split* (https://scikit-learn.org/stable/modules/generated/sklearn.model_selection.train_test_split.html, accessed on: 5 January 2022) function of scikit-learn; Notably, samples of the two classes in the training set were balanced using the data augmentation method. The training set was further divided into two different sets. The first set was used to train our models by varying their parameters, which were evaluated on the second set to optimize each AI model. We provided a statistical validation of our results by providing a ten-cross validation on our dataset that corresponded to jointly varying the training and test sets to show our model different data for each iteration (ten times) through the *cross_validate* (https://scikit-learn.org/stable/modules/generated/sklearn.model_selection.cross_validate.html, accessed on: 5 January 2022) function of scikit-learn. The best parameters for each model were reported in Table 4 to improve the reproducibility of our analysis.

Table 5 shows the results of different AI models on the test set (the 20% of a cohort of 1203 neonates) to contract a healthcare associated blood stream infection (HABSI). Notably, the high accuracy performances of LR and MLP may be used for supporting medical doctors and practitioners in their analysis. According to the statistical analysis, we found that the features, mostly affecting classification performance, were birth weight and central line catheterization days. Other most important features are gestational age and lower birthweight.

## 4. Discussion

Different surveillance studies were performed to determine how the condition of the host organism strongly connected to the onset of HAIs, which showed that patients suffering from infection rates among 6 and 40% have a higher percentage of very low birth weight neonates (birth weight ≥ 1000 g) requiring surgery [21]. Notably, the most common infections affecting neonatal patients were septicemia (45–55%), respiratory infections (16–30%) and urinary tract infections (8–18%), but the most common infections in hospitalized patients were caused by Gram-positive organisms (55.4–75%), as shown in [22]. A Carrieri et al. [23] performed a multicenter study in a cohort of 2160 newborns and revealed that 196 and 136 newborns developed late (3–10 days) and very late (>10 days) nosocomial sepsis, respectively.

A large amount of data is involved in the healthcare process or service, which were analyzed using different approaches (e.g., multicriteria decision-making [24,25,26], regression analysis [23,27,28] and advanced processing and classification techniques [13,29,30]). However, the large amount of availability of different features and their correlation pose different challenges for limiting the susceptibility to infections, which required multi-disciplinary and multi-strategic methods. Lean Six Sigma achieved promising results in healthcare [31,32,33] and the management of infections [34,35,36].

In this paper, we, designed a supervised approach using different artificial intelligence models to predict whether a patient would suffer from HABSIs under a binary task. Therefore, we first optimized each model by varying its parameters evaluated on a subset of the training set (named validation). We provided a statistical validation of our results by providing a ten-cross validation on our dataset, which corresponded to jointly varying the training and test sets to show our model different data for each iteration (ten times), and the best parameters for each model are shown in Table 4 to improve the reproducibility of our analysis. The MLP and LR better handled the imbalanced dataset (65 infected and 1038 healthy subjects), and the highest AUC and F1-macro score was observed, but SVC achieved comparable results with the other two models in accuracy.

We performed statistical analyses on a cohort of 1203 patients (65 who suffered from HABSI) to investigate risk factors for the activity of the NICU of the University Hospital “Federico II”. Notably, we emphasized that patients suffering from HABSI had a shorter gestational age and lower birthweight, which led to longer hospitalization and umbilical and central line catheterization days than non-HABSI neonates. Statistical analyses showed that significant predictors of suffering from HABSI were birth weight and central line catheterization days. In summary, the current analysis supports health practitioners in improving standards and processes to contain HAIs.

In summary, our analyses showed that birthweight and central line catheterization days were significant predictors of suffering from HABSI, and the LR and MLP achieved the highest AUC, accuracy and F1-macro score in the prediction of HABSI. Despite different efforts in the literature for the predicting the risk of HAIs and other adverse outcomes in ICU, these studies primarily focused on the use of Support Vector Machine (SVM) [16,17] or divide patients into clusters [18] without considering more recent artificial intelligence approaches (e.g., XGB or Catboost) and the large set of features used in our analysis. Our approach achieved better classification performances in accuracy and AUC w.r.t. [16,17].

Our study had some limitations due to the lack of temporal analysis. However, it did not consider the temporal features between entry into intensive areas and the first clinical evidence of infection, as shown in previous studies [23] and other types of infections.

Future works will increase the number of analyzed patients and investigate different infection types to identify possible corrective actions on the basis of the Lean Six Sigma method [34] and possible mathematical tools [37] to improve process characterization.

## 5. Conclusions

The present paper investigated risk factors related to activities in the NICU of the University Hospital “Federico II” in a cohort of 1203 patients (65 suffering from HABSI). We performed a risk analysis using logistic regression and statistical analyses to identify the main factors (birth weight and central line catheterization days) affecting neonates to contract a healthcare associated blood stream infection (HABSI). We successively designed a supervised approach using different artificial intelligence models to predict whether a patient would contracted a HABSI and found that MLP and LR better handled the imbalanced dataset (65 infected and 1038 healthy subjects), and the highest AUC and F1-macro score was observed. In summary, our analysis identified the main risk factors related to NICU activities of the University Hospital “Federico II” to jointly improve patient care of the NICU of the University Hospital “Federico II” and support practitioners in their analysis. The predictive analysis may be used in a decision support system to support practitioners in predicting and identifying hidden symptoms of healthcare-associated blood stream infections in the neonatal intensive care unit.

## Figures and Tables

**Table 1 ijerph-19-02498-t001:** Description of the meaning for parameters of Artificial Intelligence models.

Parameters	Description
C, alpha	Regularization parameter
Gamma	Kernel coefficient
Kernel	Type of kernel
n_estimators	The number of trees in the forest.
learning_rate	Learning rate schedule for weight updates.
objective	Objective function to optimize the model’s parameter
rule	Rule for splitting information
min_node_size	Minimum size of each node in the tree
hidden_layer_size	Number of neurons in hidden layer
max_iteration	Maximum number of iteration
solver	Solver for weight optimization
criterion	The function to measure the quality of a split.
max_features	The number of features to consider when looking for the best split
penalty	The norm of the penalty

**Table 2 ijerph-19-02498-t002:** Analysis of cohort under study characteristics.

	HABSIs N = 65	Non-HABSIs N = 1138	*p*-Value
Sex, boys	38 (5.71%)	628 (94.29%)	0.222
Gestational age, weeks (Median, IQR)	30 (27–33)	34 (32–37)	<0.000
Birthweight, gr (Median, IQR)	1140 (820–1470)	1940 (1442.50–2833.75)	<0.000
Length of total hospital stay, days (Median, IQR)	54 (26–83)	20 (12–33)	<0.000
Umbilical line catheterization, days (Median, IQR)	5 (0–8)	0 (0–6)	<0.000
Central line catheterization, days (Median, IQR)	14 (7–38)	0 (0–4)	<0.000

**Table 3 ijerph-19-02498-t003:** Statistical analysis for unveiling main risk factors in HABSIs infection in NICU patients.

	OR	95% CI	*p*-Value
Sex, boys	1.031	0.263–3.891	0.510
Gestational age, weeks	1.011	1.048–1.137	0.859
Birthweight, gr	0.999	0.999–1.098	0.038
Length of total hospital stay, days	1.023	0.994–1.098	0.327
Umbilical line catheterization, days	1.072	0.994–1.098	0.934
Central line catheterization, days	1.000	1.008–1.149	0.000

**Table 4 ijerph-19-02498-t004:** Best parameters for each one of the seven artificial intelligence models obtained by a ten cross validation.

AI Model	Parameters
SVC	‘C’: 1, ‘gamma’: 0.0001, ‘kernel’: ‘rbf’
CATBOOST	‘n_estimators’: 100, ‘learning_rate’: 0.01
XGB	‘learning_rate’: 0.01, ‘n_estimators ’: 100, ‘objective’: ‘binary’
RFC	‘min_node_size’: 0, ‘rule’: ‘gini’, ‘n_estimators’: 100
MLP	‘alpha’: 1e-05, ‘hidden_layer_sizes’: 14, ‘max_iter’: 1000, ‘random_state’: 1,
	‘solver’: ‘lbfgs’
RF	‘criterion’: ’entropy’, ‘max_depth’: 4, ‘max_features’: ‘auto’, ‘n_estimators’: 200
LR	‘C’: 1.0, ‘penalty’: ‘l2’

**Table 5 ijerph-19-02498-t005:** Prediction results on a dataset composed of a cohort of 1203 patients (divided into 80% for the training set and 20% for the test set) using 7 different artificial intelligence models.

AI Model	Train	Test			
	Accuracy	Accuracy	AUC	F1-Score	F1-Macro
SVC	0.9501	0.9461	0.5357	0.95	0.5527
CATBOOST	0.9438	0.9419	0.5670	0.94	0.5670
XGB	0.9428	0.9378	0.5313	0.94	0.5427
RFC	0.9469	0.9419	0.5335	0.94	0.5474
MLP	0.9511	0.9461	0.6027	0.95	0.6439
RF	0.9511	0.9419	0.5335	0.94	0.5475
LR	0.9490	0.9461	0.6027	0.95	0.6439

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
