# Peer review of "Predictive Analysis of Healthcare-Associated Blood Stream Infections in the Neonatal Intensive Care Unit Using Artificial Intelligence: A Single Center Study"

_ijerph, 2022, doi:10.3390/ijerph19052498_

Round 1

Reviewer 1 Report

A few minor changes need to be done:

  1. In line 79, expand ASA score.
  2. In Table 2, change Non-HAI to non-HABSI.
  3. Line 48 to 52, “These patients were continuously monitored by healthcare staff in all hospital ICUs and suffered from healthcare associated blood stream infection (HABSI), which were observed as possible sepsis or positive signs on blood culture in the next two days after the date of admission at the NICU on the basis of the Disease Control and Prevention directive for neonatal acute care settings.” Please make the short sentence by breaking or reframing.

4. Line 132 to 134, “According to the statistical analysis, we found that the features that most affected classification performance were birth weight and central line catheterization days, followed by gestational age and lower birthweight. ”Please reframe the sentence to make it more understandable.

Author Response

Q 1.1 A few minor changes need to be done:

In line 79, expand ASA score.

In Table 2, change Non-HAI to non-HABSI.

R 1.1 We thank the reviewer for the valuable comment. We have performed the suggested minor changes.

Q 1.2 Line 48 to 52, “These patients were continuously monitored by healthcare staff in all hospital ICUs and suffered from healthcare associated blood stream infection (HABSI), which were observed as possible sepsis or positive signs on blood culture in the next two days after the date of admission at the NICU on the basis of the Disease Control and Prevention directive for neonatal acute care settings.” Please make the short sentence by breaking or reframing.

R 1.2 Thank you for spotting this. We revised the examined sentence according to reviewer’s suggestion.

Q 1.3 Line 132 to 134, “According to the statistical analysis, we found that the features that most affected classification performance were birth weight and central line catheterization days, followed by gestational age and lower birthweight. ” Please reframe the sentence to make it more understandable.

R 1.3 Thank you for spotting this. We revised the examined sentence according to reviewer’s suggestion.

Reviewer 2 Report

None

Author Response

We thank the reviewer for the positive assessment of our work.

This manuscript is a resubmission of an earlier submission. The following is a list of the peer review reports and author responses from that submission.

Round 1

Reviewer 1 Report

The authors propose a statistical and predictive analysis about Healthcare Associated Infections in the Neonatal Intensive Care Unit in the AOU Federico II. The conclusion is supported by an extensive evaluation over 121203 patients, also using more recent approaches for predictive analysis

  The proposed approach is interesting although some points have to be improved.   In the Introduction section, there are some points/arguments that the author(s) point as given but, it is suggested, they support them with strong references (see for instance "Machine learning analysis: general features, requirements and cardiovascular applications. Minerva Cardiology and Angiology.", "A Comparison among Different Machine Learning Pretest Approaches to Predict Stress-Induced Ischemia at PET/CT Myocardial Perfusion Imaging. Computational and Mathematical Methods in Medicine, 2021). Furthermore, the authors should better describe the novelties of their approach with respect to existing ones. Furthermore, more technical details about the proposed methodology should be provided in order to improve the reproducibility of the proposed study.

Finally, I suggest to perform a linguistic revision.

Author Response

Q 1.1 The authors propose a statistical and predictive analysis about Healthcare Associated Infections in the Neonatal Intensive Care Unit in the AOU Federico II. The conclusion is supported by an extensive evaluation over 121203 patients, also using more recent approaches for predictive analysis

R 1.1 We thank the reviewer for the positive assessment of our work.

Q 1.2 The proposed approach is interesting although some points have to be improved.   In the Introduction section, there are some points/arguments that the author(s) point as given but, it is suggested, they support them with strong references (see for instance "Machine learning analysis: general features, requirements and cardiovascular applications. Minerva Cardiology and Angiology.", "A Comparison among Different Machine Learning Pretest Approaches to Predict Stress-Induced Ischemia at PET/CT Myocardial Perfusion Imaging. Computational and Mathematical Methods in Medicine, 2021).

R 1.2 We thank the reviewer for the valuable comment. We have revised the introduction section by supporting different statements through strong references.

Q 1.3 Furthermore, the authors should better describe the novelties of their approach with respect to existing ones.

R 1.3 Thank you for spotting this. We have underlined the novelties of the proposed approach at the end of the introduction section.

Q 1.4 Furthermore, more technical details about the proposed methodology should be provided in order to improve the reproducibility of the proposed study.

R 1.4 Thank you for spotting this. We have provided more details in Section 3 for improving the reproducibility of our approach.

Q 1.5 Finally, I suggest to perform a linguistic revision.

R 1.5 We thank the reviewer for all the comments and suggestions. We have conducted a comprehensive linguistic revision of the entire paper, correcting typos, grammatical errors, and overall flow and trying to improve the related readability.

Reviewer 2 Report

Please revise all the manuscript. I think it is not a definitive version of the work because some parts are short (the introduction requires additional discussion on the state of the art, hypothesis, etc.), missing (method section lacks of a lot of information), and confusing (e.g., in the result section, tables are not in the correct order). Moreover, I also noted the references are not correctly formatted (for example, line 34). 

Author Response

Q 2.2 Please revise all the manuscript. I think it is not a definitive version of the work because some parts are short (the introduction requires additional discussion on the state of the art, hypothesis, etc.), missing (method section lacks of a lot of information), and confusing (e.g., in the result section, tables are not in the correct order). Moreover, I also noted the references are not correctly formatted (for example, line 34).

R2.1 We thank the reviewer for the suggestions. We have revised the introduction section by adding information on the state of the art supported by other references. We then highlighted the main novelties of the proposed study and in particular both the statistical and predictive analysis. As for the methods section, it was improved by better describing the dataset and explaining that BSI disease is the dependent variable while the other risk factors are independent variables. We also thank the reviewer for pointing out the formatting errors. Tables and references are now correctly inserted.

Reviewer 3 Report

The authors have evaluated NICU patient population for risk factors of healthcare associated bloodstream infection (HABSI). They have also applied seven artificial intelligence tools for predictive analysis. This paper represents an effort to outline the risk factors for HABSI in NICU and utility and performance of predictive tools of artificial intelligence in NICU HABSI.

 I have the following comments and suggestions with the aim of improving the manuscript:

  1. Title: “A statistical and predictive analysis about Healthcare Associated Infections in the Neonatal Intensive Care Unit in the AOU Federico II”. May I suggest to modify to make it more representative of the study. For example “Predictive analysis of Healthcare Associated Bloodstream Infections in the Neonatal Intensive Care Unit using artificial intelligence: A single centre study

Abstract:

  1. Abstract can be more clearer, concise and informative. For example, conclusions look more informative and descriptive that results. Each section of the abstract should be representative of the similar section of the main manuscript.

Introduction:

  1. This section should be focussing on what is known and unknown in the field about this specific topic. How will this study bridge the gap? What makes this study unique?

2.

Material and Method:

  1. Please write if the data was prospectively collected or it is a retrospective data. Was the data retrieved from any database? If yes, describe the database and method of data collection.
  2. What was inclusion and exclusion criteria for the study. Authors are requested to mention them explicitly.
  3. How did you define healthcare associated BSI. How was it diagnosed and labelled?
  4. If possible use HABSI for healthcare associated bloodstream infection.
  5. Were the Apgar score and type of delivery data collected for the patients?
  6. The material and method section if describes about the place where study was carried out, it should give the details of its infection control protocols, patient to nurse ratio, etc.

7.Material and methods section mentions data collection from 2016 to 2019 but the authors while describing the data in result section write data from 2016 to 2020. Please rectify and also mention the month so that exact duration of the study can be calculated.

  1. Material and method section mentions “dependent variable (BSIs) and the different risk factors under study (i.e sex, length of total hospital stay or gestational age)” while the result section mentions “different risk factors (as dependent variables) and possible occurrence of BSI disease (as independent variable)”. Please rectify.
  2. Lines 40 to 43 and figure 1 may be replaced by more information about the infrastructural, organizational and infection control protocols and SOPs being followed in the NICU.
  3. (SVC, CATBOOST, XGB, RFC, MLP, RF and LR) these abbreviations should be used in expanded form first time in the manuscript with its designated abbreviation written in bracket.
  4. Reference no. PMID:18538699 is mentioned in the manuscript but not in reference section.
  5. Please mention the primary and secondary outcome measures being studied explicitly.
  6. Line 22 to 25 should be written in material and method section.
  7. “Furthermore, we do not consider in our analysis patients having vertical infection or other ones due to  the childbirth. ” How vertical infections and childbirth infections were ruled out?
  8. Did the authors use any severity of illness scoring system? Do the authors have mortality data of both the groups?
  9. Why Ethics Committee approval was not taken. Is there any Law of the country says that this kind of study can be done without ethics approval. Did the author apply for exemption? Please provide a clear and justified reply.

Discussion:

  1. Discussion needs to be re written fully with primary focus on artificial intelligence and machine learning utility in prediction of nosocomial infection especially HABSI in ICU (adults and pediatric) patients. It should compare the accuracy, precision, AUC, F1 score and other parameters of the models used in this study with their use and performance in other similar studies. It should also mention if the results are congruous to the findings of previous studies or it differs, if it differs what may be the reason. Authors should also maintain a good flow of discussion.

Conclusions:

  1. It should clearly present the major outcomes of the study.
  2. Future area of research or future work should be mentioned in discussion section last paragraph.

General comment:

English needs to be improved in general throughout the manuscript and flow needs to be maintained alongwith sound scientific quality of content.

Author Response

Q 3.1 The authors have evaluated NICU patient population for risk factors of healthcare associated bloodstream infection (HABSI). They have also applied seven artificial intelligence tools for predictive analysis. This paper represents an effort to outline the risk factors for HABSI in NICU and utility and performance of predictive tools of artificial intelligence in NICU HABSI.

R 3.1 We thank the reviewer for the positive assessment of our work.

Q 3.2 I have the following comments and suggestions with the aim of improving the manuscript: Title: “A statistical and predictive analysis about Healthcare Associated Infections in the Neonatal Intensive Care Unit in the AOU Federico II”. May I suggest to modify to make it more representative of the study. For example “Predictive analysis of Healthcare Associated Bloodstream Infections in the Neonatal Intensive Care Unit using artificial intelligence: A single centre study”

R 3.2 Thank you for spotting this. We have revised the paper’s title according to reviewer suggestion

Q 3.3 Abstract: Abstract can be more clearer, concise and informative. For example, conclusions look more informative and descriptive that results. Each section of the abstract should be representative of the similar section of the main manuscript.

R 3.3 We thank the reviewer for the valuable comment. We have revised the abstract by introducing the dataset characterization and improving the discussion about the obtained results.

Q 3.4 Introduction: This section should be focussing on what is known and unknown in the field about this specific topic. How will this study bridge the gap? What makes this study unique?

R 3.4 Thank you for spotting this. We have revised the introduction section to underline the main novelties of the proposed study. In particular, its novelty concerns the analysis about a cohort of $1,203$ patients, that have been investigated by both predictive and statistical analysis, stayed at the III level NCIU of the "Federico II" University Hospital in Naples from 2016 until to 2020.

Q 3.5 Material and Method: Please write if the data was prospectively collected or it is a retrospective data. Was the data retrieved from any database? If yes, describe the database and method of data collection. What was inclusion and exclusion criteria for the study. Authors are requested to mention them explicitly. How did you define healthcare associated BSI. How was it diagnosed and labelled? If possible use HABSI for healthcare associated bloodstream infection. Were the Apgar score and type of delivery data collected for the patients? The material and method section if describes about the place where study was carried out, it should give the details of its infection control protocols, patient to nurse ratio, etc.

R 3.5  We thank the reviewer for the valuable comment. The data were collected in a prospective manner by including all patients admitted to NICU through a single data source which is QuaniSDO (used for the computerization of hospital discharge forms). All hospitalized patients, from the moment of admission to discharge, have a risk record in which the epidemiologically important data for the study of hospital infections are collected.

The definition of BSI is that provided by the ECDC (European Center Disease Control), the same that is provided by the CDC in Atlanta and is based on clinical (i.e. fever) and / or microbiological (i.e. increased white blood cells, isolation of the germ)

The hospital is equipped with a manual that contains all the prevention protocols for healthcare-related infections. Each protocol has a monitoring sheet for correct use and through incidence studies it is possible to measure the effectiveness of the correct application by calculating specific indicators.

For example, the incidence rate of pneumonia associated with assisted ventilation, the incidence of urinary tract infections associated with the use of a bladder catheter.

Q 3.6 Material and methods section mentions data collection from 2016 to 2019 but the authors while describing the data in result section write data from 2016 to 2020. Please rectify and also mention the month so that exact duration of the study can be calculated.

R 3.6 Thank you for spotting this. We have revised the paper by rectifying the period of analysis, also indicating the number of months.

Q 3.7 Material and method section mentions “dependent variable (BSIs) and the different risk factors under study (i.e sex, length of total hospital stay or gestational age)” while the result section mentions “different risk factors (as dependent variables) and possible occurrence of BSI disease (as independent variable)”. Please rectify.

R 3.7 Thank you for spotting this. We have rectified into the evaluation section by indicating BSI disease as dependent variable and different risk factors as independent ones.

Q 3.8 Lines 40 to 43 and figure 1 may be replaced by more information about the infrastructural, organizational and infection control protocols and SOPs being followed in the NICU.

R 3.8 Thank you for spotting this. Every year, twice a year (November and March) the  Federico II University Hospital of Naples participates in the prevalence study of HAI organized by the ECDC which allows the comparison of the data on the European territory as well as the evaluation of its trend over time to put any critical changes highlighted. In high-risk areas, such as NICUs, incidence studies are carried out with staff who collect data in the risk records processed for each patient admitted, from admission to discharge. The data allow the preparation of reports on a quarterly basis subject to clinical audits with medical and nursing staff working in the affected areas. Training courses are organized at least three times a year for all health personnel, with compulsory attendance as part of working hours.

Q 3.9 (SVC, CATBOOST, XGB, RFC, MLP, RF and LR) these abbreviations should be used in expanded form first time in the manuscript with its designated abbreviation written in bracket.

R 3.9 Thank you for spotting this. We have used the expanded form for the abbreviation in their first use.

Q 3.10 Reference no. PMID:18538699 is mentioned in the manuscript but not in reference section.

R 3.10 Thank you for spotting this. We have added the required reference.

Q 3.11 Please mention the primary and secondary outcome measures being studied explicitly.

R 3.11 We thank the reviewer for the valuable comment. We have revised the entire paper underlying that our aims are to provide a statistical and predictive analysis about HBSI; in particular, in Section 3 we describe the obtained results, whose outcome has been discussed in Section 4. Summarizing:

  1. We performed a risk analysis on the basis of logistic regresson and statistical analysis to unveil the main factors affecting neonates to contracts a blood stream HAIs (HABSI). We underline that that patients suffering of a HABSI have gestional age and lower birthweight shorter leading to a longer hospitalization and umbilical line catheterization days and central line catheterization days longer than non-HABSIs neonates. It is further possible to show that significant predictors of suffering from HABSIs are birth weight and central line catheterization days.
  2. We have designed a supervised approach using different artificial intelligence models to predict if a patient has contracted the HABSI. In this analysis, it is easy to note how MLP and LR better handle the imbalanced dataset (65 infected and 1038 healthy), observing highest AUC and F1-Macro score, although SVC achieves comparable results w.r.t. the other two models in terms of accuracy.

   Q 3.12 Line 22 to 25 should be written in material and method section. “Furthermore, we do not consider in our analysis patients having vertical infection or other ones due to the childbirth. ” How vertical infections and childbirth infections were ruled out?

R 3.12 Thank you for spotting this. In our study we do not consider childbirth-related or vertically transmitted infections because women who give birth are subjected to vaginal and rectal swabs for the research of bacteria related to them while the search for viruses is done by blood.

Q 3.13 Did the authors use any severity of illness scoring system? Do the authors have mortality data of both the groups?

R 3.13 Thank you for spotting this. Patients studied were not stratified by disease severity although the database used reports the ASA score. We have mortality data for both groups but they were not included in the processed dataset

Q 3.14 Why Ethics Committee approval was not taken. Is there any Law of the country says that this kind of study can be done without ethics approval. Did the author apply for exemption? Please provide a clear and justified reply.

R 3.14 We thank the reviewer for the valuable comment. In compliance with the Declaration of Helsinki and with the Italian Legislative Decree 211/2003, Implementation of the 2001/20/CE directive, since no patients/children were involved in the study, the signed informed consent form and the ethical approval are not mandatory for these type of studies. Furthermore, in compliance with the regulations of the Italian National Institute of Health, our study is not reported among those needing assessment by the Ethical Committee of the Italian National Institute of Health. 

Q 3.15 Discussion: Discussion needs to be re written fully with primary focus on artificial intelligence and machine learning utility in prediction of nosocomial infection especially HABSI in ICU (adults and pediatric) patients. It should compare the accuracy, precision, AUC, F1 score and other parameters of the models used in this study with their use and performance in other similar studies. It should also mention if the results are congruous to the findings of previous studies or it differs, if it differs what may be the reason. Authors should also maintain a good flow of discussion.

R 3.15 We thank the reviewer for the valuable comment. We have revised the discussion section according to the reviewer’s suggestion; we have firstly discussed the obtained results obtained by using Artificial Intelligence approach, also discussing optimization phase for each AI model. Furthermore, we investigated the statistical analysis and its main outcome.

Q 3.16 Conclusions: It should clearly present the major outcomes of the study. Future area of research or future work should be mentioned in discussion section last paragraph.

R 3.16 Thank you for spotting this. We improved the Conclusions section by underlying the main outcomes of our statistical and predictive analysis.

Q 3.17 General comment: English needs to be improved in general throughout the manuscript and flow needs to be maintained along with sound scientific quality of content.

R 3.17 Thank you for spotting this. conducted a comprehensive linguistic revision of the entire paper, correcting typos, grammatical errors, and overall flow and trying to improve the related readability.

Round 2

Reviewer 2 Report

This is an improved version of a manuscript aiming to develop a model for predicting the risk of Healthcare Associated Infections in the Neonatal Intensive. However, I have some points to be considered:

Introduction: Please describe if there are some models for predicting the risk of HAIs and other adverse outcomes in ICU (please consider the following: doi: 10.1016/j.jhin.2021.02.025    doi: 10.1016/j.jhin.2020.09.030    doi: 10.3390/jcm10050992) 

Methods: Please describe how the Authors have managed the low number of events in their dataset. How the two datasets (training and test) are obtained? Have the authors compared these datasets? 

Did the authors evaluate overfitting?

Could the authors provide more information about the weight of each variable in the model with the best perfomance?

Discussion: What is the clinical utility of these findings?

Author Response

Q 2.1 This is an improved version of a manuscript aiming to develop a model for predicting the risk of Healthcare Associated Infections in the Neonatal Intensive.

R 2.1 We thank the reviewer for the positive assessment of our approach.

Q 2.2 However, I have some points to be considered: Introduction: Please describe if there are some models for predicting the risk of HAIs and other adverse outcomes in ICU (please consider the following: doi: 10.1016/j.jhin.2021.02.025    doi: 10.1016/j.jhin.2020.09.030    doi: 10.3390/jcm10050992)

R 2.2 We thank the reviewer for the valuable comment. We have improved the introduction section by adding the suggested paper and discussing the main novelties of the proposed analysis w.r.t. the suggested ones.

Q 2.3 Methods: Please describe how the Authors have managed the low number of events in their dataset. How the two datasets (training and test) are obtained? Have the authors compared these datasets?

R 2.3 Thank you for spotting this. We have described in Section 3.2 how we have divided our dataset in train (80%) and test (20%) by using a stratified strategy in order to consider the same unbalanced proportion both in training and test sets; in particular, the samples of two classes in the training set has been balanced by using data augmentation method. Furthermore, the training set has been, further, divided into two different sets, where the first one has been used to train our models varying their parameters, that have been evaluated on the second one in order to optimize each AI model. Furthermore, we provide a statistical validation of our results by providing a ten-cross validation on our dataset, that corresponding to jointly varying train and test set for showing to our model different data for each iteration (ten times).

Q 2.4 Did the authors evaluate overfitting?

R 2.4 Thank you for spotting this. We have performed ten cross validations during training phase in order to avoid the overfitting issue as well as evaluate in terms of F1-score in order to investigate if the majority class affects classification performances (as shown in Table 5).

Q 2.5 Could the authors provide more information about the weight of each variable in the model with the best perfomance?

R 2.5 Thank you for spotting this. According to the statistical analysis, we note that the features that most affect classification performance are birth weight and central line catheterization days, following by the gestational age and lower birthweight shorter. We provide further details at the end of Predictive analysis section.

Q 2.6 Discussion: What is the clinical utility of these findings?

R 2.6 We thank the reviewer for the valuable comment. We have improved the conclusion by discussing main clinical utility of the obtained results. In particular, in our analysis, we are interesting in identifying the main risk factors related to the activity of the NICU of the University Hospital "Federico II" in order to jointly improve patient care of the NICU of the University Hospital "Federico II" and supporting practitioners in their analysis. Furthermore, the predictive analysis could be used in a decision support system that can support practitioners in predicting and identifying hidden symptoms of Healthcare Associated Bloodstream Infections in the Neonatal Intensive Care Unit.

Reviewer 3 Report

  1. The term described should be healthcare associated blood stream infection (HABSI) and not bloodstream HAI as observed in the manuscript.

  1. “The definition of BSI is that provided by the ECDC (European Center Disease Control), the same that is provided by the CDC in Atlanta and is based on clinical (i.e. fever) and / or microbiological (i.e. increased white blood cells, isolation of the germ). ” The authors need to write it explicitly in the text.
  2. How vertical infections and childbirth infections were ruled out? In reply to this, the authors have written “In our study we do not consider childbirth-related or vertically transmitted infections because women who give birth are subjected to vaginal and rectal swabs for the research of bacteria related to them while the search for viruses is done by blood.” This answer is not well understood.
  3. Discussion needs to be further improved. Please also compare and contrast your study with other similar (adult or pediatric ) studies.
  4. Conclusions: It should be clear and concise and not too much explanatory.
  5. Overall the flow and english language needs to be greatly improvised.

Author Response

Q 3.1 The term described should be healthcare associated blood stream infection (HABSI) and not bloodstream HAI as observed in the manuscript.

R 3.1 We thank the reviewer for the valuable comment. We have revised the entire paper in order to use the correct concept.

Q 3.2 “The definition of BSI is that provided by the ECDC (European Center Disease Control), the same that is provided by the CDC in Atlanta and is based on clinical (i.e. fever) and / or microbiological (i.e. increased white blood cells, isolation of the germ). ” The authors need to write it explicitly in the text.

R 3.2 We thank the reviewer for spotting this. We have explicitly stated the discussed sentence in the Introduction section (see row 29-31).

Q 3.3 How vertical infections and childbirth infections were ruled out? In reply to this, the authors have written “In our study we do not consider childbirth-related or vertically transmitted infections because women who give birth are subjected to vaginal and rectal swabs for the research of bacteria related to them while the search for viruses is done by blood.” This answer is not well understood.

R 3.3 Thank you for spotting this. We have improved the sentence by adding more details about the viral infection and the reason behind the feature selection.

Q 3.4 Discussion needs to be further improved. Please also compare and contrast your study with other similar (adult or pediatric ) studies.

R 3.4 We thank the reviewer for the valuable comments. We have improved our discussion compared our results with respect to different state-of-the-art approaches for predicting the risk of HAIs and other adverse outcomes in ICU (see references [1,2,3]).

Q 3.5 Conclusions: It should be clear and concise and not too much explanatory.

R 3.5 Thank you for spotting this. We have shortened our conclusion section, also adding few sentences about the main clinical utility of the obtained results.

Q 3.6 Overall the flow and english language needs to be greatly improvised.

R 3.6 We thank the reviewer for all the comments and suggestions. We have conducted a comprehensive linguistic revision of the entire paper, correcting typos, grammatical errors, and overall flow and trying to improve the related readability.

References

[1] Barchitta, Martina and Maugeri, Andrea and Favara, Giuliana and Riela, Paolo Marco and Gallo, Giovanni and Mura, Ida and Agodi, Antonella and Network, SPIN-UTI (2021). A machine learning approach to predict healthcare-associated infections at intensive care unit admission: findings from the SPIN-UTI project. Journal of Hospital Infection, 112, 77-86.

[2] Barchitta, M., Maugeri, A., Favara, G., Riela, P. M., Gallo, G., Mura, I., & Agodi, A. (2021). Early prediction of seven-day mortality in intensive care unit using a machine learning model: results from the SPIN-uti project. Journal of clinical medicine, 10(5), 992.

[3] M. Barchitta and A. Maugeri and G. Favara and P.M. Riela and C. {La Mastra} and M.C. {La Rosa} and R. Magnano {San Lio} and G. Gallo and I. Mura and A. Agodi and Marco Brusaferro and Salesia Fenaroli and Ennio Sicoli and Maria Teresa Montagna and Raffaele Squeri and Rosario Massimo {Di Bartolo} and Salvatore Tribastoni and Anna Rita Mattaliano and Patrizia Bellocchi and Giacomo Castiglione and Marinella Astuto and Anna Maria Longhitano and Maria Concetta Monea and Giorgio Scrofani and Antonino {Di Benedetto} and Maria Carmela Riggio and Giuseppe Manta and Romano Tetamo and Ignazio Dei and Irene Pandiani and Antonino Cannistrà and Paola Piotti and Massimo Girardis and Elena Righi and Pierangelo Sarchi and Luca Arnoldo and Silvio Brusaferro and Salvatore Coniglio and Albino Borracino and Sergio Pintaudi and Massimo Minerva and Marina Milazzo and Emanuela Bissolo and Alberto Rigo and Leila Fabiani and Franco Marinangeli and Paolo Stefanini and Marcello Mario D'Errico and Abele Donati and Stefano Tardivo and Francesca Moretti and Alberto Carli and Riccardo Pagliarulo and Aida Bianco and Maria Pavia and Marcello Pasculli and Cesare Vittori and Giovanni Battista Orsi and Cristina Arrigoni and Patrizia Laurenti and Franco Ingala and Patrizia Farruggia (2021). Cluster analysis identifies patients at risk of catheter-associated urinary tract infections in intensive care units: Findings from the SPIN-UTI Network. Journal of Hospital Infection, 107, 57-63.